# COVID-19 Vaccine Coverage and Potential Drivers of Vaccine Uptake among Healthcare Workers in SOMALIA: A Cross-Sectional Study

**DOI:** 10.3390/vaccines10071116

**Published:** 2022-07-13

**Authors:** Hassan Abdullahi Dahie, Jamal Hassan Mohamoud, Mohamed Hussein Adam, Bashiru Garba, Najib Isse Dirie, Maryan Abdullahi Sh. Nur, Fartun Yasin Mohamed

**Affiliations:** 1Nursing and Midwifery Departments, Faculty of Medicine and Health Sciences, SIMAD University, Mogadishu 2526, Somalia; dahie@simad.edu.so; 2Department of Public Health, Faculty of Medicine and Health Sciences, SIMAD University, Mogadishu 2526, Somalia; ganowyare@simad.edu.so; 3Institute for Medical Research, SIMAD University, Mogadishu 2526, Somalia; garba.bashiru@udusok.edu.ng; 4Department of Veterinary Public Health and Preventive Medicine, Faculty of Veterinary Medicine, Usmanu Danfodiyo University, Sokoto 2346, Nigeria; 5Department of Urology, Dr. Sumait Hospital, Faculty of Medicine and Health Sciences, SIMAD University, Mogadishu 2526, Somalia; drnajib@simad.edu.so; 6Department of Obstetrics and Gynecology, Dr. Sumait Hospital, Faculty of Medicine and Health Sciences, SIMAD University, Mogadishu 2526, Somalia; dr.maryan.abdullahi@gmail.com; 7Departments Microbiology and Medical Laboratory Sciences, Faculty of Medicine and Health Sciences, SIMAD University, Mogadishu 2526, Somalia; fartunyasin141@simad.edu.so

**Keywords:** COVID-19 vaccine, COVID-19, healthcare workers, vaccine uptake, Somalia

## Abstract

Healthcare workers (HCWs) are one of the most vulnerable groups for contracting COVID-19 and dying as a result of it. Over 10,000 HCWs in Africa have been infected with COVID-19, according to the World Health Organization, making it a substantial occupational health threat for HCWs. To that end, Somalia’s Ministry of Health has ordered that all healthcare personnel obtain the COVID-19 vaccination to safeguard themselves and the community they serve. In this investigation, we aimed to assess the COVID-19 vaccination coverage and its associated factors among healthcare workers in Somalia. A cross-sectional study was employed to examine COVID-19 vaccination coverage among healthcare personnel in Somalia. The data were obtained via an online questionnaire supplied by Google forms between December 2021 and February 2022, where a total of 1281 healthcare workers from the various federal states of Somalia were recruited. A multinomial regression analysis was used to analyse the factors associated with COVID-19 vaccine uptake. Overall, 1281 HCWs participated (630 females, 651 males) with a mean age and standard deviation of 27.7 years ± 7.1. The overall vaccine coverage was 37.4%. Sex, age, the state of residency, education level, specialization, hospital COVID-19 policy, vaccine availability at the centre, COVID-19 treatment centre, and health facility level were the factors that influenced the COVID-19 vaccine uptake among health professionals in Somalia. Male healthcare employees were 2.2 times (odds ratio-OR = 2.2; confidence interval-CI: 1.70, 2.75, *p* < 0.001) more likely than female healthcare workers to be fully vaccinated. The survey discovered that the COVID-19 vaccine coverage among health professionals was quite low, with the major contributing factors being accessibility, security challenges and literary prowess. Additional efforts to enhance vaccination uptake are needed to improve the COVID-19 vaccination coverage.

## 1. Introduction

The ongoing COVID-19 pandemic has continued to wreck national economies, global health, and the livelihood of people all over the globe. The disease has maintained an upward trend since its inception in terms of morbidities and mortalities with the highest number of confirmed cases recorded in January 2022 [1]. According to the WHO COVID-19 dashboard (16 May 2022), a total of 521,694,216 confirmed cases and 6,274,111 deaths have been reported globally [1]. Unfortunately, the burden of the disease is projected to continue to rise driven by community transmission by asymptomatic individuals [2,3]. The initial efforts to stem the tide and flatten the number of confirmed COVID-19 cases and death included both therapeutic and nontherapeutic measures [4,5,6]. Other nonpharmaceutical interventions that were found to be very effective at reducing the spread of SARS-CoV-2 viruses are the wearing of facemasks, hand hygiene, and physical distancing [7,8,9]. However, over time these measures have become tranquil requiring better sustainable approaches, particularly among resource-limited countries where access to facemasks and disinfectants could be a problem in addition to the mere impossibility of avoiding crowds and gatherings due to religious, social, and cultural peculiarities [10].

To break the chain of COVID-19 transmission, herd immunity, which is an indirect protection, must be conferred on a sufficiently large proportion of the global population. However, achieving herd immunity from natural infection and recovery followed by the development of immunity could be counterproductive because of the unprecedented strain the disease could exert on the already scarce healthcare resources [11]. Therefore, vaccination is considered the most suitable and effective intervention for attaining herd immunity and controlling the ongoing pandemic. However, to achieve this goal, around 80–90% of the population must acquire COVID-19 immunity, either through prior infection or vaccination [12]. Unfortunately, the global inequities in terms of access to COVID-19 vaccines, and other factors associated with COVID-19 vaccine uptake and acceptability, are seen as a major threat to this success [12,13,14].

In the context of Somalia, the first COVID-19 case was reported on 16 March 2020 [15]. This was then followed by community transmission that was reportedly instigated by the first-line staff of the Ministry of Health from contact with people arriving from overseas. Since the confirmation of the first case, the Somali government has taken proactive steps to halt the spread of COVID-19 which included suspension of both local and international flights, closure of public institutions, and forming the COVID-19 task force to enforce and monitor adherence to the set-out preventive measures in the country. Moreover, following the receipt of its first consignment of COVID-19 vaccines through the COVAX facility in March 2021, the Somali government decided to prioritize an estimated 300,000 frontline workers, the elderly, and people with chronic health conditions [16]. The decision to prioritize these essential workers and higher-risk individuals was to ensure that healthcare and other essential services continued to function at maximum capacity and that deaths among the elderly and people with comorbidities were reduced significantly.

The rapid development and deployment of the COVID-19 vaccines globally have been adjudged an unprecedented achievement, however, it has also fuelled vaccine hesitancy among both healthcare workers and the general public [17]. According to the WHO’s Strategic Advisory Group of Experts on immunization, “Vaccine hesitancy” is a behavioural response exhibited by members of the public that ranges from a lack of trust in vaccines or vaccine manufacturers to a lack of sufficient knowledge about the success of vaccines in preventing many deadly diseases before COVID-19, as well as accessibility issues [18]. Despite the above-mentioned factors, the literature materials about COVID-19 vaccination uptake and hesitancy are still scarce. As of 9 May 2022, Somalia has administered a total of 2,677,716 doses of the COVID-19 vaccine (AstraZeneca, Sinopharm, and Johnson & Johnson) which represents a paltry 8.6% of the population among which we do not know how many are healthcare workers.

Given the lack of information in Somalia regarding factors affecting the uptake of COVID-19 vaccines, and the attitudes of frontline healthcare workers (HCWs), we undertook this study to survey the Somali healthcare workers to determine the vaccination coverage since the vaccines were deployed and to understand factors that affect the vaccine uptake to propose solutions, particularly because HCWs are at elevated risk of contracting the infection, they play a critical role in providing healthcare services to the nation, and they can strongly influence the vaccine uptake among their patients [19].

## 2. Methods and Materials

### 2.1. Study Design and Setting

This study was undertaken using a cross-sectional online survey to investigate COVID-19 vaccine coverage among healthcare workers across all the federal member states of Somalia and the potential drivers of vaccine hesitancy. The study was conducted over a three-month (4 December 2021 to 10 February 2022) period, where all members of the healthcare service were invited to participate in the anonymous online survey. The WHO behavioural and social drivers (BeSD) model was adapted for COVID-19 vaccination among healthcare workers with slight modifications.

### 2.2. Participants and Sampling Method

All healthcare workers aged 18 years and above living in Somalia and working in a healthcare setting regardless of patient care contact and role in healthcare settings were eligible to participate in the study. A cluster sampling was primarily utilized to categorize the study area into seven different regions (comprising the six Federal States and Benadir Regional Administration). The individuals in each stratum (Federal State) were recruited using a convenient nonprobability sampling technique. Informed consent was obtained prior to enrolment in the study. Incomplete responses were excluded from the analysis.

### 2.3. Sample Size Determination

Assuming the healthcare workers of Somalia to be 19,306 according to the Ministry of Health and Social Services [20,21]. Since there is no prior similar study about the COVID-19 vaccine in Somalia, we took (*p*) as 50% to achieve the maximum sample size for the current study and a margin of error of 3% (95% CI: 47–53%), we calculated a sample size of 1067 individuals. By adding an 8% nonresponse rate, and a design effect of (1.1), the minimum required sample size becomes 1267.6 ≈ 1268. However, the present study received 1305 responses (Figure 1). We screened the data and excluded 24 responses due to partial or incomplete information. Finally, we found a total of 1281 responses for the final analysis.

### 2.4. Data Collection Tools

An online English questionnaire was created using a Google form. The survey guide was a modification of the WHO BeSD model of vaccine uptake for healthcare workers. The survey tool was distributed through specific social media platforms to various HCW groups as well as administrative officers at major healthcare institutions within the Benadir Regional Administration and the six Somali federal states.

### 2.5. Measures/Variables

The questionnaire included questions on demographic information (age, gender, education level, the state of primary residence in the last six months, and the specialty of the healthcare worker). On the other hand, the questionnaire also contained questions about the features and services of the health centre relating to COVID-19 including, “Does your hospital/health centre offer COVID-19 treatment/care?” (Answer options: Yes/No); “Is COVID-19 vaccine available at your facility?” (Answer options: Yes/No); “What is the level of your healthcare facility?” (Answer options: “Primary Level, Secondary Level, and Tertiary Level”; “Is it mandatory for any medical staff to take COVID-19 vaccine in your health facility?” (Answer options: Yes/No). Similarly, inquiries about the health status of the workers were also made using questions such as: “Do you have hypertension?” (Answer options: Yes/No); “Are you diabetic?” (Answer options: Yes/No); “Do you have Asthma?” (Answer options: Yes/No); “Do you have Cardiovascular diseases?” (Answer options: Yes/No). Furthermore, the participants were asked questions relating to their COVID-19 status using the following phrases: Have you ever checked your COVID-19 status? (Answer options: Yes/No); where they answered “YES” a follow-up question “What was the test result?” was also asked.

Vaccination coverage was determined by asking: “Have you taken COVID-19 vaccines?” (Answer options: Yes/No); if “YES”, a follow-up question, “What was the type of the vaccine? Oxford-AstraZeneca, Pfizer-BioNTech, Moderna, Johnson & Johnson, I can’t recall, Not Applicable”: where they answered “YES” for taking the vaccines, they were asked to state “How many doses have you taken? 1 dose, 2 doses, or not applicable.” When the respondents answered “NO” for not taking the COVID-19 vaccine, they were asked “Can you give the reason(s) why?” (Answer options: Fear of the vaccine’s adverse effects, due to unavailability of COVID-19 vaccines in our area, due to COVID-19 vaccines not being accessible, I believe that the vaccine is not effective, I already had COVID-19, so I think I am immune to the disease).

### 2.6. Statistical Analysis and Management Tools

After the questions and variables were evaluated for completeness, consistency, and accuracy, the data were cleaned, coded, entered, and analysed using the statistical package for social sciences (SPSS) version 25.0 software. The main outcome variables of interest were whether respondents had been vaccinated or not, as well as the reasons for not being vaccinated. For qualitative variables, descriptive statistics such as relative (percentage) and absolute frequencies were used, whilst quantitative variables were reported using mean and standard deviation. An odds ratio (OR) analysis using a multinomial logistic regression analysis model was also employed to determine any independent relationships with COVID-19 vaccination coverage. All significance tests were two-tailed, with statistical significance set at *p* < 0.05.

## 3. Results

After screening and filtering the responses from the 1305 HCWs, and removal of partial or incompletely filled questionnaires (24), a total of 1281 responses were retained and analysed.

Data relating to the demographic characteristics of the health care employees are presented in Table 1. The results showed that a slight majority of the HCWs were females (50.8%). The mean age for the respondents was 27.7 years, with a standard deviation of 7.1 years. Concerning their level of education and specialization, the results revealed that the majority had a bachelor’s degree (74.2%), while 5.3%, 11.5%, and 9.1% had a certificate, diploma, and postgraduate degrees, respectively. In terms of the area of specialization, we observed that 31% of the HCWs studied were nurses, 21%, 12%, 12%, and 11% were medical doctors, public health officials, midwives, and lab technicians, respectively. Furthermore, based on the state distribution, the majority of the HCWs were found to be working in different health institutions within the Benadir region (23%), with 18%, 16.5%, 14.1%, 12.8%, 11.1%, and 4.6% employed in Somaliland, Hirshabelle state, Jubaland state, the Southwest state, Puntland state, and Galmudug state, respectively.

According to the features of the various health facilities in terms of COVID-19-related healthcare (Table 2), the results showed that 65.9% of the healthcare institutions did not offer COVID-19 treatment and care and that only 27.9% of the health institutions were offering COVID-19 vaccination for members of the public during the study period. We also observed that 53.5% of the respondents were working for secondary-level health facilities across the various states and the Benadir region. This investigation also observed that the majority (63.9%) of the healthcare facilities had an existing policy requiring their staff to be vaccinated against COVID-19. Moreover, 61.7% of the HCWs reported that they had checked their COVID-19 status with 14% indicating their result turned out to be positive.

Out of 48.6% of health care workers that claimed they have been vaccinated against COVID-19, 40.8% received the Oxford-AstraZeneca vaccine, 6.7% received Johnson & Johnson, while only 0.5% indicated that they received Moderna. However, a paltry 8 (0.6%) HCWs reported that they could not recall the type of COVID-19 vaccine they took.

Regarding the comorbidities status of healthcare workers (Table 3), asthma was the most prevalent comorbidity, with 6.7% of healthcare professionals having the condition, followed by diabetes (4.9%), hypertension (4.4%), and cardiovascular diseases (3.2%).

The study found that around 38% of Somali healthcare professionals were fully vaccinated against COVID-19, while 11% were partially vaccinated (Figure 2). However, more than half (51%) of HCWs did not receive any COVID-19 vaccination, even though they were at the forefront of the pandemic’s battle. HCWs are considered fully vaccinated two weeks after their second dose in a two-dose series, such as the Pfizer-BioNTech and Moderna vaccines, or two weeks after the single-dose Janssen vaccine [22].

Concerning the distribution of vaccination coverage within the states of Somalia, the study observed that the state with the highest proportion of vaccinated healthcare workers during the study period was Puntland in which (49.3%) of its health workers received the COVID-19 vaccines, followed by Somaliland (40.9%), Jubaland (39%), the Southwest (37.8%), the Benadir region (35.7%), Hirshabelle (31.1%), and Galmudug (20.3%), (Figure 3).

One of the primary goals of the current study was to identify the factors that drive the HCWs’ hesitation to receive the COVID-19 vaccination. As depicted in Figure 4, more than a half (50.6%) of the 653 health care workers who reported to have not received any COVID-19 vaccination cited a fear of the vaccine’s side effects, followed by 16% who questioned the vaccine efficacy, 11.4% claimed the unavailability of the vaccine, while a small proportion (5.6%) reported challenges in vaccine accessibility. Furthermore, the argument of 17% of the unvaccinated HCWs was based on believing that they developed immunity against the COVID-19 disease as they already had the infection, implying that vaccination was unnecessary.

Table 4 below shows the results of the multinomial logistic regression model that estimates the odds ratio of a healthcare professional being partially vaccinated or fully vaccinated versus not vaccinated. After controlling for other covariates, the characteristic features of the HCWs that were most likely to be associated with a greater odd when comparing full or partial vaccination, versus no vaccination were: gender, age, federal state, level of education, and health specialty. Similarly, being a COVID-19 vaccination and treatment centre, the level of the health centre and mandatory COVID-19 vaccination status of health workers were also determined.

When comparing male health workers to their female counterparts, the males were found to have 2.2 times higher chances of being fully vaccinated compared to those not being vaccinated (OR = 2.2, 95% CI = 1.703–2.751), and a 1.7 times higher chance of being partially vaccinated compared to not being vaccinated (OR = 1.3, 95% CI = 1.152–2.374). Similarly, health workers whose ages were between 36 and 45 years compared to those aged 25 years or below were found to be 2.3 times more likely to be fully vaccinated compared to those not being vaccinated (OR = 2.3, 95% CI = 1.445–3.756) and about 2.4 times higher chance of being partially vaccinated compared to nonvaccinated (OR = 2.4, 95% CI = 1.285–4.4428).

With respect to the various federal states, health professionals from Hirshabelle state when compared to those from the Southwest state had four times higher chances of being fully vaccinated compared to not being vaccinated (OR = 4.1, 95% CI = 2.356–7.242) and almost five times higher chance of being partially vaccinated compared to unvaccinated (OR = 4.8, 95% CI = 2.186–10.664). Likewise, those healthcare professionals who had postgraduate degrees compared to those at the certificate level were found to have a 4.3 times higher possibility of taking the full COVID-19 vaccination compared to nonvaccinated (OR = 4.3, 95% CI 2.153–8.437) and 1.8 times higher chances of being partially vaccinated compared to nonvaccinated (OR = 1.8, 95% CI = 0.557–6.079). Furthermore, concerning the specialty of the health workers, nurses compared to midwives had 1.6 times higher chances of being fully vaccinated compared to nonvaccinated (OR = 1.6, 95% CI = 1.070–2.467) and almost two times higher chance of being partially vaccinated compared to unvaccinated (OR = 1.8, 95% CI = 1.003–3.359).

Health workers working at COVID-19 treatment centres compared to those working at non-COVID-19 health centres had 2 times more chances of being fully vaccinated compared to unvaccinated (OR = 2.0, 95% CI = 1.291–2.140) and 1.7 times more of being partially vaccinated relative to unvaccinated (OR = 1.7, 95% CI = 1.164–2.571). Moreover, in respect of COVID-19 vaccine availability, health workers whose centres had the COVID-19 vaccine relative to those who did not have, had 2 times higher chances of being fully vaccinated compared to unvaccinated (OR = 2.0, 95% CI = 2.628–4.736) and 2.4 times higher chances of being partially vaccinated compared to unvaccinated (OR = 2.4, 95% CI = 1.580–3.743).

Regarding the level of the health facility, health workers working for tertiary hospitals compared to those working at the primary level had 2.2 times higher chances of being fully vaccinated compared to nonvaccinated (OR = 2.2, 95% CI = 1.421–3.261) and 1.4 times higher chances of being partially vaccinated compared to nonvaccinated (OR = 1.4, 95% CI = 0.784–2.625) (Table 5).

## 4. Discussion

This study assessed COVID-19 vaccination coverage and factors driving hesitancy among health professionals across regions of Somalia. The study found that 624 (48.7%) of the healthcare workers were vaccinated for COVID-19 at least once, while 37.4% of them were fully vaccinated. This was lower than coverages in developed countries such as the United States (83.3%) [23] and developing countries such as Ethiopia (62.1%) [24]. One of the major contributors to vaccine hesitancy in Somalia and indeed most countries has been misinformation driven by social media platforms that the COVID-19 vaccines tend to make one infertile [25,26]. This may have been one of the major impediments to achieving wider coverage especially because of pride attributed to births in Somalia. Considering the critical role the HCWs have and their reputation and influence in their respective communities, it is very necessary that efforts be geared towards some form of target group-specific education about the importance and safety of this vaccine including the long-term benefits to promote vaccination readiness and acceptance.

Other demographic variables studied were age, sex, educational level, the area of specialization of the HCWs, as well as the states where they reside and work during the ongoing pandemic. In this regard, male healthcare workers were found to have increased odds of accepting the COVID-19 vaccination (OR = 2.2; CI: 1.70, 2.75, *p* < 0.001) compared to their female counterparts. This finding is in line with studies earlier reported in Ghana and Ethiopia where gender was found to be a significant determinant of vaccine acceptance. The results from this study also support initial trends pre-COVID-19 era which also indicates a systematic gender difference during studies of vaccination coverage among adolescents including higher vaccination rates among men than women in the case of influenza and pandemic influenza vaccinations [27,28]. Other reasons behind this apparent vaccination gender gap point to geographical, social, and gender disparities that continue to exist in Somalia where women tend to have lower access to mobility and information on where to get vaccinated. Moreover, there is a disproportionate rate of employment between males and females within the Somali healthcare system which is attributed mainly to men having higher work motivation [29]. Additionally, we also found that the odds of being fully vaccinated compared to unvaccinated was increased among older ages (26–35 years, 36–45 years, 46+) compared to the reference category (25 years and below) (OR = 1.8, 95% CI = 1.402–2.322, *p* < 0.001), (OR = 2.3, 95% CI = 1.5–2.8, *p* < 0.001), (OR = 2.8, 95% CI = 1.375–5.645, *p* = 0.004), respectively. This finding may not be unconnected with the global policy of prioritizing older adults due to their higher risk of having severe COVID-19 infection [30,31]. Other studies conducted in the Kingdom of Saudi Arabia showed that older males were more likely to be willing to accept being vaccinated against COVID-19.

As shown in Table 5, full vaccination coverage was found to be higher among healthcare workers with bachelor’s and postgraduate degree levels compared to the reference group (healthcare workers with certificates) (OR  =  1.9; 95% CI = 1.110–3.490, *p ≤* 0.001) and (OR = 4.3, 95% CI = 2.153–8.437, *p* ≤ 0.001) respectively. Similarly, Alhassan et al. [32] found that healthcare workers with higher educational qualifications were more likely to receive the COVID-19 vaccine than those with lower educational qualifications, implying that limited access to information may perpetuate misconceptions about the COVID-19 vaccine and impede its uptake. However, there was no significant difference in vaccination status between healthcare workers with a certificate and those with a diploma.

On the other hand, healthcare professionals from Hirshabelle state compared to those from the Southwest state had a four times higher likelihood of being fully vaccinated compared to not accepting the vaccination (OR = 4.1, 95% CI = 2.356–7.242) and almost five times higher chance of being partially vaccinated compared to not vaccinated (OR = 4.8, 95% CI = 2.186–10.664). This could be attributed to the fact that the health authorities in Hirshabelle made it mandatory for all healthcare workers in the state to take the COVID-19 vaccination. Furthermore, this research focused on major cities such as Jowhar, Beledweyn, and Bulaburte, where the majority of medical workers were situated and where internet connectivity was available Challenges to equitable COVID-19 vaccine access in conflict-affected locations have been attributed to vaccine procurement and distribution, among others [33]. These may have been the contributing factors affecting vaccine coverage in the areas occupied by the insurgents in Somalia. Reports have shown that scarcity of resources, challenges related to logistic support, competing priorities, and insecurity are some of the major drivers limiting safe access to populations living behind conflict lines [34].

According to the study, health workers whose hospitals mandated their staff to take the COVID-19 vaccines compared to those whose hospitals did not mandate had a 14.2 times higher likelihood of being fully vaccinated compared to those not vaccinated (OR = 14.2, 95% CI = 10.638–18.696) and 11.9 times higher chance of being partially vaccinated (OR = 11.9, 95% CI = 7.804–18.235). The study findings are consistent with multisociety statements on COVID-19 vaccination as a condition of employment for healthcare personnel [35]. The decision by many organizations to make it mandatory for their employees to take the COVID-19 vaccine was done in a bid to make their workplace safer, and protect other employees, their families, and the community as a whole. This was because outbreaks of COVID-19 were initially more associated with social venues such as restaurants, and religious organizations, as well as workplaces.

Although this study found no statistically significant difference in the likelihood of receiving the COVID-19 vaccine between healthcare workers with and without chronic diseases, a study conducted in Ethiopia found that participants with a history of chronic disease were more likely to accept the vaccine [36]. This can be attributed to the fact that the bulk of the health care providers surveyed were young people with no history of chronic illnesses.

To the best of our knowledge, this is the first study to investigate the coverage of the COVID-19 vaccine and related characteristics among Somali health professionals involving the six federal states of Somalia and the Benadir Regional Administration. Furthermore, a selection bias may also have existed as the methods may have discriminated against older adults and those with poor internet access. Finally, the questionnaire was self-administered which suggests the possibility of having information bias. However, piloting the questionnaire and revising it for clarity and content before publishing the final version should have minimized the risk of misunderstanding the questions.

## 5. Conclusions

From the findings of this study, we can see that institutional policy mandating staff to be vaccinated, access to quality information common among HCWs with bachelor and postgraduate degrees, and incidence of chronic disease/comorbidity were the major factors encouraging uptake of the COVID-19 vaccines. Similarly, accessibility to the vaccine which was affected by the security challenges facing the country was also found to be a significant factor that affected the rate of vaccine uptake among the healthcare workers residing in some conflict-prone states. Even though the Somali government intended for healthcare workers to be the first cadres to receive the COVID-19 vaccine as they are the first to come into contact with patients who have COVID-19 infections, the survey discovered that the COVID-19 vaccine coverage among health professionals was quite low (38%). Health workforces are one of the critical components of the building blocks of the reforming health systems in Somalia. They are a reliable source of health information, and their uptake of COVID-19 vaccines can influence the wider public. With respect to the distribution of vaccination coverage within the states of Somalia, the study revealed that Puntland, Somaliland, Jubaland, the Benadir region, the Southwest, Hirshabelle, and Galmudug were 49.3%, 40.9%, 39%, 37.8%, 35.7%, 31.1%, and 20.3%, respectively. Concerning the factors influencing COVID-19 vaccine uptake, sex, age, education level, specialization, hospital COVID-19 policy, vaccine availability, being a COVID-19 treatment centre, and health facility level were all found to be drivers of COVID-19 vaccine uptake among health professionals in Somalia.

Hence the need for authorities at the federal and member state ministries of health, in partnership with other stakeholders to consider instituting mandatory vaccination for health professionals and developing vaccine delivery plans.

## Figures and Tables

**Figure 1 vaccines-10-01116-f001:**
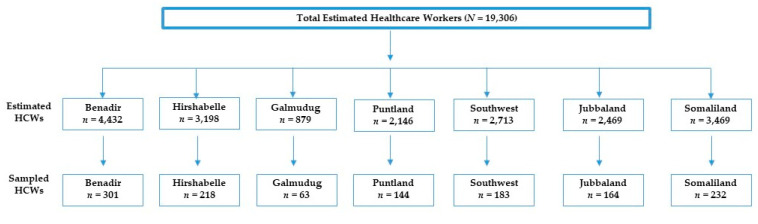
The proportionate fraction of the sample size from each Somali federal member states (2022).

**Figure 2 vaccines-10-01116-f002:**
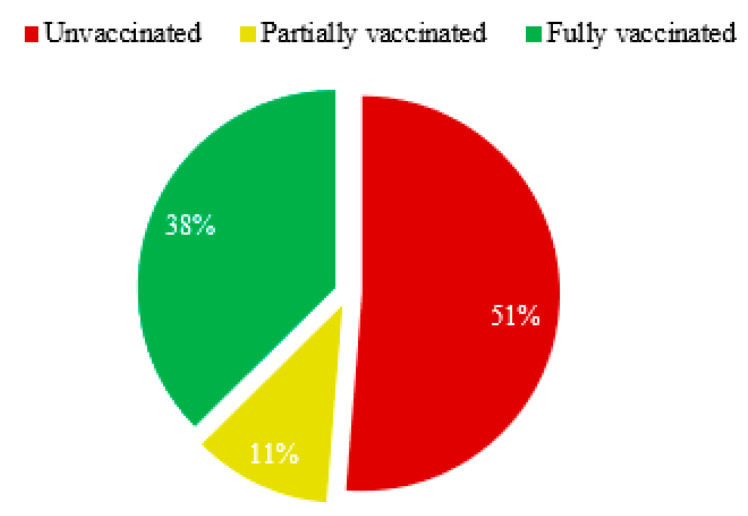
Overall COVID-19 Vaccination coverage.

**Figure 3 vaccines-10-01116-f003:**
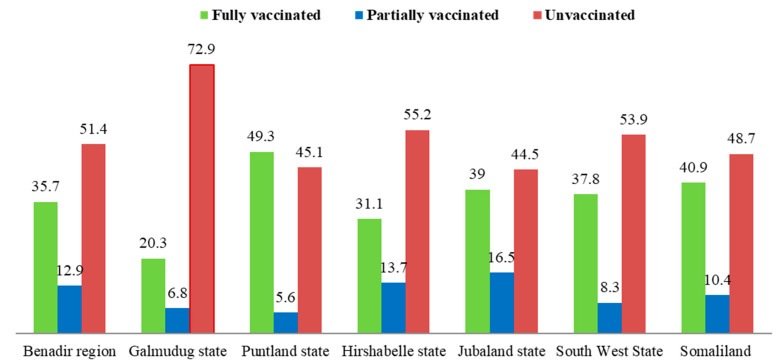
COVID-19 vaccination coverage by federal states.

**Figure 4 vaccines-10-01116-f004:**
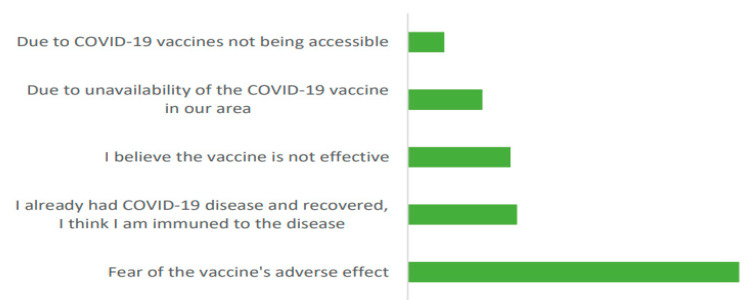
The reasons given by the unvaccinated HCWs for not taking vaccines (*n* = 653).

**Table 1 vaccines-10-01116-t001:** Demographic characteristics among healthcare workers.

Demographic Characteristics	No. Participants	Percentage (%)
**Gender**		
Male	651	50.8
Female	630	49.2
**Age of respondents**		
25 and below	608	47.5
26–35 years	537	41.9
36–45 years	97	7.9
46 and above	39	3
**Educational level**		
Certificate	68	5.3
Diploma	147	11.5
Bachelor	950	74.2
Postgraduate	116	9.1
**Health Specialties**		
Nursing	398	31.1
Midwifery	154	12
Lab technicians	141	11
Medical doctor	265	20.7
Public health official	159	12.4
Dentist	13	1
Pharmacist	33	2.6
Community health worker	42	3.3
Clinical officer	12	0.9
Nutritionist	15	1.2
Other	49	3.8
**Province/Geographical area**		
Benadir region	294	23
Hirshabele state	212	16.5
Puntland state	142	11.1
Galmudug state	59	4.6
Jubaland state	164	12.8
Southwest state	180	14.1
Somaliland	230	18

**Table 2 vaccines-10-01116-t002:** Health centres’ characteristics.

Variable	No. Participants	Percentage (%)
**Does your hospital/health centre offer COVID-19 treatment/care?**		
Yes	437	34.1
No	844	65.9
**Is the COVID-19 vaccine available at your facility?**		
Yes	358	27.9
No	923	72.1
**What is the level of your centre?**		
Tertiary level	137	10.7
Secondary level	689	53.5
Primary level	455	35.5
**Is it mandatory for any medical staff to take the COVID-19 vaccine in your health facility?**		
Yes	819	63.9
No	462	36.1
**Have you checked your COVID-19 Status?**		
Yes	790	61.7
No	491	38.3
**If YES, what was the test result?**		
Positive	179	14
Negative	611	47.7
NA	491	38.3
**Have you taken COVID-19 Vaccines?**		
Yes	623	48.6
No	658	51.4
**If YES, what was the type of vaccine?**		
Oxford-AstraZeneca	523	40.8
Moderna	6	0.5
Johnson & Johnson	86	6.7
I can’t recall	8	0.6
NA	658	51.4
**If YES, how many doses have you taken?**		
1 dose	230	18
2 doses	393	30.6
NA	658	51.4

**Table 3 vaccines-10-01116-t003:** Comorbidities status.

Variable	No. Participants	Percentage (%)
**Hypertension status**		
Yes	57	4.4
No	1224	95.6
**Diabetes status**		
Yes	63	4.9
No	1218	95.1
**Asthma status**		
Yes	86	6.7
No	1195	93.3
**Cardiovascular disease status**		
Yes	41	3.2
No	1240	96.8

**Table 4 vaccines-10-01116-t004:** Demographic characteristics, comorbidities, and COVID-19 vaccination status.

Demographic Characteristics	Partially Vaccinated			Fully Vaccinated		
	OR	95% CI	*p* Value	OR	95% CI	*p* Value
**Gender**						
Male	1.7	1.152–2.374	0.006 *	2.2	1.703–2.751	0.001 *
Female	Ref			Ref		
**Age of respondents**						
25 and below	Ref			Ref		
26–35 years	0.9	0.606–1.341	0.901	0.9	0.606–1.341	0.901
36–45 years	2.4	1.285–4.4428	0.006 *	2.4	1.285–4.4428	0.006 *
46 and above	1.7	0.595–4.875	0.321	1.7	0.595–4.875	0.321
**Geographical region**						
Southwest	Ref			Ref		
Benadir	1.9	0.998–3.976	0.051	1.8	1.158–2.902	0.010 *
Galmudug	4.1	1.822–9.273	0.001 *	3.9	2.267–7.038	0.001 *
Hirshabelle	4.8	2.186–10.664	0.0001 *	4.1	2.356–7.242	0.001 *
Jubaland	1.6	0.687–3.936	0.264	3.1	1.831–5.301	0.001 *
Puntland	1.1	0.420–3.104	0.796	4	2.377–6.949	0.001 *
Somaliland	2.1	0.945–4.568	0.069	3.1	1.887–5.171	0.001 *
**Educational level**						
Certificate	Ref			Ref		
Diploma	1.8	0.638–5.309	0.259	1.4	0.738–2.781	0.288
Bachelor	2.1	0.842–5.576	0.109	1.9	1.110–3.490	0.021 *
Postgraduate	1.9	0.612–0.6355	0.256	4.1	2.125–8.252	0.001 *
**Health Specialties**						
Midwifery	Ref			Ref		
Nursing	1.8	1.003–3.359	0.049 *	1.6	1.070–2.467	0.023 *
Lab technician	1.3	0.596–2.706	0.536	1.5	0.888–2.441	0.134
Medical doctor	1.2	0.602–2.331	0.625	1.8	1.174–2.826	0.007 *
Public health	1.04	0.479–2.265	0.919	1.7	1.099–2.898	0.019 *
Dentist	0.6	0.070–4.909	0.623	0.4	0.090–2.033	0.285
Pharmacist	0.3	0.032–2.026	0.197	0.8	0.357–1.955	0.679
CHWs	0.7	0.206–2.853	0.692	1.5	0.715–3.089	0.289
Clinical officer	-	-	-	2.9	0.899–9.952	0.074
Nutritionist	0.9	0.097–7.287	0.874	2.0	0.706–6.464	0.179
Other	1.9	0.716–5.357	0.191	2.2	1.058–4.314	0.034 *
**Comorbidities status**						
**Hypertension status**						
Yes	1.9	0.548–6.782	0.307	0.6	0.292–1.2300	0.163
No	Ref			Ref		
**Diabetic status**						
Yes	0.56	0.229–1.389	0.213	1.6	0.772–3.284	0.207
No	Ref			Ref		
**Asthma status**						
Yes	1.4	0.572–3.340	0.473	1.2	0.663–1.997	0.617
No	Ref			Ref		
**Cardiovascular diseases**						
Yes	0.6	0.197–1.913	0.400	1.1	0.464–2.427	0.887
No	Ref			Ref		

* Indicate the statistical significance.

**Table 5 vaccines-10-01116-t005:** Health centre characteristics and COVID-19 vaccination status.

Demographic Characteristics		Partially Vaccinated			Fully Vaccinated		*p*-Value
		OR	95% CI	*p* Value	OR	95% CI	
**Treatment centre for COVID-19**							
Yes		1.7	1.164–2.571	0.007 *	2.0	1.291–2.140	0.001 *
No		Ref			Ref		
**Availability of COVID-19 vaccine**							
Yes		2.4	1.580–3.743	0.001 *	3.5	2.628–4.736	0.001 *
No		Ref			Ref		
**Level of health centre**							
Primary level		Ref			Ref		
Secondary level		0.8	0.571–1.232	0.370	1.4	1.076–1.807	0.012 *
Tertiary level		1.4	0.784–2.625	0.241	2.2	1.421–3.261	0.001 *
**Mandatory for medical staff to take COVID-19**							
Yes		11.9	7.804–18.235	0.001 *	14.2	10.638–18.696	0.001 *
No		Ref			Ref		

* Indicate the statistical significance.

## Data Availability

The data described in this work can be obtained from the corresponding author upon request. Due to participant privacy concerns, the data is not publicly available.

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
