# Peer review of "COVID-19 Vaccine Coverage and Potential Drivers of Vaccine Uptake among Healthcare Workers in SOMALIA: A Cross-Sectional Study"

_vaccines, 2022, doi:10.3390/vaccines10071116_

Round 1

Reviewer 1 Report

Reviewer comments to the Authors -

Regarding the article titled “COVID-19 vaccine coverage and potential drivers of vaccine uptake among healthcare workers in Somalia: A cross-sectional study”.

There are the following prominent opinions :

1.     Generally speaking, the conception is problematic; the authors can rely on vaccine centers' information more than the online google sheet.

2.     According to the title of this paper, potential drivers among health care workers are not clearly described in this paper.

3.     The primary purpose of this paper is to warn/inform the somalin health sector to enhance the immunization program among the health workers in Somali. They were NOT founding or illustrating a current clinical problem in covid 19 vaccination process.

4.     WHO clarifies Herd immunity, also known as 'population immunity.' It is indirect protection from an infectious disease when a population is immune either through vaccination or immunity developed through the previous infection". Therefore, herd immunity is related to the Somali population, not health workers, because they work among the patients.

5.     In the discussion, the section needs to discuss your data and current scientific references; however, the author included a lot of imaginary concepts more than scientific ones. (ex- "healthcare workers from states with relative peace, like Somaliland & Puntland, had a greater likelihood of being fully vaccinated than those health workers in politically turbulent states”). Overall, this manuscript looks just survey results on the vaccination value in health care works in Somali. There are no scientific findings and meaning. 

In addition, there are some minor comments :

1) Provide DOI links of references and modify the format of the references by referring to other articles in this journal.

2) English language use (syntax, grammar, etc.) and details must be substantially improved to make the manuscript acceptable.

3) There are many apparent errors in the article, some special words need to be italicized, and the first occurrence in the article needs to be spelled out.

Date Sent:

05-31-2022

Reviewer 2 Report

Comments to the Author
The manuscript: COVID-19 vaccine coverage and potential drivers of vaccine uptake among healthcare workers in Somalia: A cross-sectional study. I suggest accepting this manuscript after the revision, and the authors should consider the suggestions described below:
The paper needs to be improved in the following manners, which are necessary to be answered before further processing

1)    Authors should carefully revise and corrected all the grammatical issues and follow the scientific norms in the whole manuscript

2)    The resolution of Figures and recheck

3) Please use updated and recent papers in the literature review to give more sense to the reader.

4) Conclusions could be more specific and to the point.

5) Please more elaborate on the novel aspect of your work at the end of the introduction.

   6)  In the Reference part, please add references from recent years.

Ultra-Precise Label-Free Nanosensor Based on Integrated Graphene with Au Nanostars Toward Direct Detection of IgG Antibodies of SARS-CoV-2 in Blood

Ultra-sensitive viral glycoprotein detection NanoSystem toward accurate tracing SARS-CoV-2 in biological/non-biological media

Ultra-Precise Label-Free Nanosensor Based on Integrated Graphene with Au Nanostars Toward Direct Detection of IgG Antibodies of SARS-CoV-2 in Blood

In silico designing a candidate vaccine against breast cancer

Recent Biotechnological Approaches for Treatment of Novel COVID-19: From Bench to Clinical Trial

Ultra-Precise Label-Free Nanosensor Based on Integrated Graphene with Au Nanostars Toward Direct Detection of IgG Antibodies of SARS-CoV-2 in Blood

Bioinorganic Synthesis of Polyrhodanine Stabilized Fe3O4/Graphene Oxide in microbial supernatant media for anticancer and antibacterial applications

Green synthesis of silver nanoparticles toward bio and medical applications: review study

Development of clay nanoparticles toward bio and medical applications

Recent advancements in polythiophene-based materials and their biomedical, geno sensor and DNA detection

Separation of Ni (II) from Industrial Wastewater by Kombucha Scoby as a Colony Consisted from Bacteria and Yeast: Kinetic and Equilibrium Studies

Introduction of magnetic and supermagnetic nanoparticles in new approach of targeting drug delivery and cancer therapy application

Reviewer 3 Report

I thoroughly appreciated this article and the clear, easy writing style of the authors.   I wish that other scientific papers were written so clearly.    I endorse publication.   A couple of minor points for the authors’ consideration:

·         The term, “fully vaccinated”, at least in the United States, seems to depend on who is speaking and when they are speaking.   Does fully vaccinated include a booster shot?  A second booster shot?   My point is simple.  I would like the authors to insert a brief definition of “fully vaccinated”  around where they mention the concept on page 7.

·         While not necessary for this article, I would be interested in a cross tabulation of health facility mandate for worker vaccination by whether the worker actually got vaccinated.   Do the mandates work?

·         Accessibility of vaccines would seem to be an important factor, and while I would not require this analysis, it would be interesting to run an analysis of compliance  limiting to health care workers who did not indicate problems with access.    This would allow you to separate the cognitive and emotional factors from the supply chain factors in hesitancy.   I am concerned that their inclusion in the analysis may lead to an overestimate of hesitancy.   How many would have taken the vaccine if it were available?  Again, this is not necessary for publication, but it would be of interest.  

·         Question – 265 respondents were reported to be physicians.   Only 116 respondents were reported to have postgraduate degrees.  Is this because Somalia, like Ireland and UK, award physicians with bachelor’s degree after graduating medical school?   I understand that some countries, unlike my home country of United States, allow entry directly into a 5 year medical school program and award a bachelor’s degree.  

Reviewer 4 Report

Besides the importance of thematic for the people of Somalia that I fully respect, I consider that the issue is not relevant for the global science describing a very particular reality.  Some minor comments: 1) The brief conclusion in abstract is missing. Please add a sentence or two which emphasize the impact/meaning of this manuscript. 2) Please explain what “OR” and “CI” stand for. 3) In English commas are used to separate numbers greater than 999. Thus such values as “1305” or „1281” should be written „1,305” and „1,281”. (page 3) 4) It is not clear what exactly statistical test were used. Also please add information about post hoc test if used.

Round 2

Reviewer 1 Report

1. According to the covid situation in Somalia, admissible difficulties in recording liable information, however publishing in international recognized journal need convincing, vulnerable data for future perspective. As a suggestion, the author can be based on the WHO Somalia database, or major hospital vaccination process database depending on how many vaccines are received in Somalia and so on; with online google questionaries for health care workers.

2. This manuscript still lacks clinical problem findings. 

3. Still English correction needed. 

4. The reviewer cannot find any scientific or vaccination problems in this manuscript. 

Author Response

Dear reviewer,

We thank you once again for your interest and objective criticism of our work. Many of the observations made have definitely improved the manuscript, and we thank you sincerely for that. We’ve also tried to address the additional issues raised in this second round of review to the best of our ability and understanding. Below are the responses (blue ink) to the additional concerns

Reviewer: 1

Area

Comment to the author

Authors’ response

Methodology

According to the covid situation in Somalia, admissible difficulties for recording liable information however publishing in international recognized journal need convincing, vulnerable data for future perspective. As a suggestion, the author can be based on the WHO Somalia database or major hospital vaccination process database depending on how many vaccines are received in Somalia and so on; with online google questionaries for health care workers.

Dear respected reviewer, we appreciate your concerns regarding some aspects of our manuscript. As reported in the article, our intention was not to determine the vaccine uptake rate in Somalia (I assume that is what you are referring to when you request that we consult WHO database or major hospitals about vaccines received). What we essential seek to understand is the reasons behind the decision of some of the HCWs not to take the vaccine. We however, acknowledge how important information from the WHO database would’ve benefited us. Thank you

General

I am not satisfied with your answer.  There is no specific clinical problem finding.

Dear respected reviewer, I am really struggling to understand what you mean by “clinical problem”. In my little understanding, clinical problems associated with vaccines and vaccination could be failure for vaccine to render protection after administration, adverse health effects following vaccination (like the blood clotting reported or other allergic reactions), etc. Recall that we made reference to reports that only 8.6% of the Somali population has received COVID-19 vaccination (line 95-99). Despite we don’t know the number of health workers covered, this still represent a significant problem that has to be tackled. And to do that we need to understand why the uptake is low, which is the main objective of this study. In this regard we were able to demonstrate that the drivers of vaccine uptake (which we found to be access to quality information, mandate for vaccination at work place, chronic disease) and low uptake which we found security challenge to be among the major contributors.

Still need correction.

We have also rechecked to correct the errors

Round 3

Reviewer 1 Report

This manuscript much improved.